# Consumption of Sinlek Rice Drink Improved Red Cell Indices in Anemic Elderly Subjects

**DOI:** 10.3390/molecules26206285

**Published:** 2021-10-17

**Authors:** Peerasak Lerttrakarnnon, Winthana Kusirisin, Pimpisid Koonyosying, Ben Flemming, Niramon Utama-ang, Suthat Fucharoen, Somdet Srichairatanakool

**Affiliations:** 1Department of Family Medicine, Faculty of Medicine, Chiang Mai University, Chiang Mai 50200, Thailand; peerasak.lerttrakarn@cmu.ac.th (P.L.); wkusiris@gmail.com (W.K.); 2Oxidative Stress Cluster, Department of Biochemistry, Faculty of Medicine, Chiang Mai University, Chiang Mai 50200, Thailand; pimpisid_m@hotmail.com (P.K.); benf9900@gmail.com (B.F.); 3Department of Earth and Environment, Faculty of Science and Engineering, School of Natural Sciences, University of Manchester, Manchester M13 9PT, UK; 4Cluster of High-Value Products from Thai Rice and Plants for Health, Faculty of Agro-Industry, Chiang Mai University, Chiang Mai 50200, Thailand; niramon.u@cmu.ac.th; 5Department of Product Development Technology, Faculty of Agro-Industry, Chiang Mai University, Chiang Mai 50200, Thailand; 6Thalassemia Research Center, Institute of Molecular Biosciences, Mahidol University Salaya Campus, Nakornpathom 71300, Thailand; suthat.fuc@mahidol.ac.th

**Keywords:** anemia, cognition, elder, hemoglobin, iron, *Oryza sativa*, rice

## Abstract

Iron fortifications are used for the treatment of iron-deficiency anemia; however, iron dosing may cause oxidative damage to the gut lumen. Thai Sinlek rice is abundant in iron and contains phytochemicals. We aimed at evaluating the effect of an iron-rice (IR) hydrolysate drink (100 mL/serving) on neurological function, red cell indices and iron status in elders. Healthy elderly subjects were divided into three non-anemic groups and one anemic group. The non-anemic groups consumed one WR (2 mg iron/serving) and two IR drinks (15 and 27 mg iron/serving) (groups A, B and D, respectively), while the anemic group consumed one IR drink (15 mg iron serving) (group C) every day for 30 days. There were no significant differences in the MMSE Thai 2002 and PHQ9 test scores for members of all groups, while the nutrition scores and body weight values of group D subjects were significantly increased. Hemoglobin (Hb) and mean corpuscular hemoglobin concentrations increased significantly only in group C. Serum iron and transferrin saturation levels tended to increase in group A, while these levels were decreased in members of group C. Serum antioxidant activity levels were increased in all groups, and were highest in group C. Thus, consumption of an IR drink for 15 days functioned to increase Hb and antioxidant capacity levels in anemic elders.

## 1. Introduction

Anemia is a worldwide public health problem that is caused by a range of factors, including inherited disease, acute blood loss, iron deficiency, end-stage renal disease, pregnancy and aging [1]. Iron-deficiency anemia (IDA) is a common nutritional disease suffered by 1 billion people and must be treated with iron formulations (such as ferrous sulphate, ferric carboxymaltose and Fe_3_O_4_@Astragalus polysaccharide nanoparticles) [2,3,4]. Though iron supplementation is an effective therapy, gastrointestinal disturbance and oxidative bowel damage are known to be uncompliant side effects. IDA is diagnosed by measuring biomarkers of iron stores, blood iron parameters and red cell indices. Nutritional anemia in the elderly (about one-third of all those diagnosed with the disease) is important to address because the mortality risk has significantly increased. An adequate energy and protein diet, along with effective iron and vitamin supplementation, have been acknowledged in the prevention and treatment of certain diseases such as anemia [5]. Incidences of iron deficiency have been reported at considerable levels in elderly populations, of which females appear to be at higher risk than males. However, elderly populations of both genders are known to have a similar low dietary iron intake of 10–11 mg/day [6].

Meng and colleagues have reported on the iron content found in different forms of rice (*Oryza sativa* L.), including black rice, red rice, sticky rice and rice millet. A higher degree of iron content was found in black rice when compared with other rice varieties [7]. A previous study has revealed that red blood cell numbers were increased in early weaned piglets after they were fed with high-iron rice [8]. Along with the establishment of iron-fortified foods, the process of agronomic biofortification of rice grains has been increasingly developed in major rice-producing countries for the purposes of addressing micronutrient malnutrition in human populations [9]. Rice beverages are value-added and manufactured by means of enzymatic hydrolysis and bacterial fermentation [10,11,12]. The effects of temperature, reaction time, raw materials-to-water weight ratio and suitable α-amylase enzyme hydrolysis need to be determined in order to produce high yields of nutrients and bioactive compounds, while sustaining low carbohydrate content in the soluble or concentrated products [12]. However, oxidative mucosal toxicity of ferrous sulphate tablets was found when deglutition disorders were present in elderly patients. Consequently, appropriate iron pharmaceutical formulations (such as syrups and drinks) should be provided to these IDA patients [13]. Red rice grain is concentrated with iron and many phytochemicals (such as phenolics, proanthocyanidins, oryzanol and vitamin E) that are known to exert beneficial effects on human health [14,15,16,17]. Hypothetically, Sinlek rice hydrolysate that is abundant in iron together with phytochemicals should be considered a potential nutrient in the treatment of IDA patients. In this study, we focused on evaluating the effects of iron-rice drinks on iron status, red cell indices and brain functions in Thai anemic elderly patients.

## 2. Results

### 2.1. Subject Information

Firstly, one hundred and thirty-four elderly volunteers applied for enrolment in this study and twenty-six individuals were excluded. From the relevant calculation, there should be twenty-seven subjects in non-anemic groups A, B and D and in anemic group C at the beginning of the study. However, two subjects in groups A, B and C were excluded due to the occurrence of diseases or related complications. Furthermore, two additional anemic subjects in group C were withdrawn during the course of this study. As is shown in Table 1, a total of one-hundred elderly subjects were enrolled in the study and these subjects were divided into four groups. These groups were then identified as non-anemic groups A (*n* = 25), B (*n* = 25) and D (*n* = 27), and an anemic group C (*n* = 23). The number of female subjects in the anemic group C was lower than in groups B and D, with statistical significance (*p* < 0.05). In addition, the anemic subjects were found to be older than the non-anemic subjects (*p* < 0.05). The average ages of the subjects in groups C, A, B and D were 74.3 ± 7.2, 68.5 ± 5.7, 71.6 ± 7.7 and 68.9 ± 7.9 years respectively, and their height values were 153.4 ± 8.2, 154.1 ± 7.2, 152.1 ± 8.0 and 150.8 ± 6.2 cm, respectively. However, before the study, marital status, education, profession, health behaviors, including exercise, smoking and alcohol drinking, body mass index (BMI) and co-morbidities including chronic diseases, hypertension, diabetic mellitus and hyperlipidemia that were reported in the members of these four groups were not determined to be significantly different among all four groups (Table 1 and Table 2).

Moreover, Barthel activities of daily living (ADL), Mini Mental State Examination (MMSE Thai 2002) and Patient Health Questionnaire 9 (PHQ9) scores were found to be non-significantly different in groups A and B, and between the anemic group and the non-anemic group (Figure 1). The ADL scores for groups A, B, D and C were 19.84 ± 0.80, 19.84 ± 0.37, 19.89 ± 0.42 and 19.83 ± 0.49 points, respectively.

### 2.2. Health, Nutritional Scores and Neurological Function

Maintaining and retarding the decline of physical and cognitive function, along with achieving the optimal degree of control of chronic diseases in aging populations, is an important global goal for researchers in the hopes of increasing the life expectancy of humans. Iron supplementation and therapeutic interventions are a nutritional strategy that could be used to prevent IDA and other iron-related disorders in the elderly. As is shown in Figure 2, there were no significant differences in the MMSE Thai 2002 and PHQ9 test scores for members of all groups before and after the study. Surprisingly, the nutrition scores and body weight (BW) values of the subjects in group D were significantly increased from 12.74 ± 0.76 to 13.30 ± 1.03 (^a^*p* < 0.05) and 59.50 ± 12.47 kg to 60.37 ± 13.29 kg (^b^*p* < 0.05) respectively, during the course of this study. Iron is essential for increase of erythrocyte mass, ribonucleotide reductase-catalyzed cell proliferation and tissue expansion [18]; inversely, primary negative effects associated with IDA include deficits in body weight gain. For instance, pigs fed with a 450–600 mg ferrous sulfate-supplemented basal diet showed significant increases of BW, SI and TIBC values when compared with those fed with the basal diet, suggesting that the iron fortification should improve growth performance [19]. Some studies have reported that excess body mass or obesity was associated with iron excess [20]. Increased iron availability and iron fortification positively affect human growth, and increased growth in humans provided greater amounts of iron 18. In the present study, there was greater BW gain in group D with the IR drink (27 mg iron/serving) than that in group B with the IR drink (15 mg iron/serving) and group A with the WR drink (2 mg iron/serving), possibly due to increased proliferation and expansion of cells (such as adipocytes). Moreover, there were no significant differences in systolic blood pressure (SBP) as well as diastolic blood pressure (DBP) in this study.

### 2.3. Hematopoietic Activity

Changes in hematological parameter values for 1, 2 and 3 visits for all groups are shown in Figure 3. Mean blood hemoglobin (Hb) levels of non-anemic subjects in group B were found to have decreased from 13.50 ± 0.92 to 12.97 ± 0.95 g/dL and to 12.83 ± 0.75 g/dL (*p* < 0.05) after receiving the low-dose IR drink for 15 and 30 days, respectively. Remarkably, the Hb level in group C increased from 1 to 2 visits (11.71 ± 0.71 to 11.96 ± 0.95 g/dL) and decreased after 3 visits (11.86 ± 0.86 g/dL). Mean hematocrit (Hct) values of non-anemic subjects in groups B and D were found to have decreased and were statistically significant. Significant differences were recorded in the mean Hct levels in groups B and D, mean white blood cells (WBC) in groups A and D, mean polymorphonuclear cells (PMN) in groups A and B, mean lymphocyte values in group A, mean eosinophil values in groups A, B and D, mean monocyte values in group D, the average mean corpuscular volume (MCV) in group C, the average mean corpuscular hemoglobin (MCH) values in groups A, B and D and the mean platelets in group D. However, no statistically significant differences were observed in the mean RBC values for all groups. Mean changes in Hct values in group A (0.35% ± 0.42%) were found to reveal statistically significant differences when compared with those of group B (−0.72% ± 0.39%) in the last 15 days (−0.07% ± 0.15% and −0.67% ± 0.12%). Using the same low-dose IR drink, the mean change of Hb level variables in group C was found to have increased with statistically significant differences when compared with group B in the first 15 days (0.24 ± 0.13 and −0.53 ± 0.11 g/dL) and 30 days (0.14 ± 0.11 and −0.67 ± 0.12 g/dL). This may be indicative of the effect of the IR drink in increasing Hb levels in anemic subjects when compared to non-anemic subjects. In addition, mean changes in the values of MCV in group C (1.28 ± 0.72 fL) were significantly different when compared with those in groups A, B and D (−0.78 ± 0.39, −2.08 ± 0.87 and −2.75 ± 2.60 fL, respectively) over 30 days. Mean changes in the values of MCH in group C (−0.06 ± 0.14 pg) were significantly different from those in groups A, B and D (−0.17 ± 0.15, −0.77 ± 0.28 and −0.62 ± 0.15 pg, respectively) over 30 days. Changes in the values of mean corpuscular hemoglobin concentration (MCHC) level variables in group C (0.60 ± 0.21 g/dL) revealed significant differences when compared with those of group B (−0.50 ± 0.29 g/dL) in the first 15 days of this study. Nonetheless, mean changes in values of WBC, monocyte and platelets (PLT) were not found to reveal statistically significant differences for all groups.

### 2.4. Iron Status

The levels of serum Ft concentrations were significantly varied in the subjects of groups A, B and C; however, the Ft levels were not changed by the administration of rice drinks in all groups during the course of this study. Evidently, serum iron (SI) and total iron-binding capacity (TIBC) levels were normal and non-significantly different in all groups, even among anemic subjects (group C). Furthermore, the subjects were not observed to be affected by the IR drinks. Moreover, the administration of rice drinks was found to influence the levels of SI and TIBC significantly in subjects in groups B and D. The rice drink was also found to significantly influence the serum Ft levels of non-anemic subjects, but not in the anemic subjects (Figure 4).

### 2.5. Serum Antioxidant Capacity

Antioxidant capacity (AC) levels before and after the administration of the rice drink increased significantly in all groups, and the changes in groups A, B, D and C after the administration for 30 days were 31 ± 20, 47 ± 17, 38 ± 13 and 71 ± 19 μg TE/mL, respectively (Figure 5). This finding suggests that the effects of the anti-oxidative phytochemical ingredients were most effective in the anemic elders.

### 2.6. Blood Biochemical Parameters

The values of all serum biochemical parameters for all groups were generally at the normal levels before and after the study (Table 3, Table 4 and Table 5). The levels of serum lipids, total protein (TP), albumin (Alb) and FBS were not found to be significantly different when comparisons were made between non-anemic and anemic groups (Table 3). The consumption of WR and IR drinks was found not to influence the levels of these parameters. The levels of blood urea nitrogen (BUN) and serum creatinine (CRE) in anemic subjects seemed to be higher than in non-anemic subjects, but the differences were not determined to be significantly different (Table 4). Similarly, the levels of serum uric acid (UA), sodium ion (Na^+^) and potassium ion (K^+^) were not significantly different among the groups and were found to be unchanged during the rice drink intervention experiments (Table 4). Likewise, the levels of serum TP, Alb and globulin (Glo) were not significantly different among the groups and during the period of intervention. In contrast, the levels of serum alanine aminotransferase (ALT) and aspartate aminotransferase (AST) activities were different among the groups, for which the levels tended to decrease non-significantly in all groups during the course of the intervention (Table 5).

## 3. Discussion

With regard to a global IDA, rice fortification could increase the intake of micronutrients, particularly iron. The World Health Organization (WHO) has recommended to fortify polished rice with iron to achieve intake that meets the average requirement (150–300 g rice/cap/day (7 mg iron/100 g)) of adults [21]. The iron fortification of rice is a promising strategy that can be applied to improve iron nutrition in order to treat and prevent iron deficiency. With regard to iron bioavailability, fractional iron absorption in young women was found to be significantly higher in subjects who consumed ferric pyrophosphate (FPP) and citrate-fortified rice (4 mg iron/meal) when compared with those who consumed FPP-fortified rice and a citrate solution. However, no differences were observed in those who consumed the ferrous sulfate (FS)-fortified rice reference meal [22]. In Thailand, Chitpan and coworkers had previously fortified dried broken rice with ferrous sulfate, ferrous lactate or ferric ethylenediaminetetraacetic acid (FeEDTA) at 5.3 mg of iron/20 g serving. It was found that FeEDTA fortification was the most suitable option and was revealed to be the most stable over time [23]. In addition, the fortification of FeEDTA (5–10 mg of iron/day) in phyto-diets containing phytates and polyphenolics did not produce any direct toxic effects but did increase zinc and copper absorption [24]. Nonetheless, iron fortification must follow the WHO guidelines in order to achieve the highest degree of iron bioavailability and efficacy [25]. In this study, we used cross-bred Sinlek rice in the iron-rice drink (27 mg of iron/100 mL serving) that contained anti-oxidative phenolic compounds and γ-oryzanol. Importantly, this specific iron-rice drink received a high degree of acceptance among tested consumers. Vitamin C is an important factor in the enhancement of iron bioavailability [26,27], so it has to be taken into consideration for oral iron intervention. A previous study has reported original vitamin C levels of 29.7 and 36.6 mg in 100 g of wet Riceberry and Sinlek rice grains, respectively (Oraphan Srichuenchom B.Sc. Special Problem Study, Kasetsart University 2014). Herein, we have detected very low vitamin C contents in all rice drinks, which was probably due to the losses incurred during hydrolysis and heat lability. The limitations of this study include small serum volumes and an absence of determination of vitamin C concentrations in the sera. Vitamin C is abundant in citrus fruit and functions to convert dietary Fe^3+^ to more soluble Fe^2+^, which can be readily absorbed into the intestinal epithelial membrane via the divalent metal iron transporter 1. With this in mind, commercial citric acid, which is a weak tridentate iron (Fe^2+^ and Fe^3+^)-chelating chelator [28,29], was added as a food-flavoring and iron-stabilizing agent to the drinks. For the satisfaction of consumers, 0.1% citric acid chelator (food grade) was added as an ingredient to soya sauces that had been fortified with ferrous sulfate, sodium FeEDTA, ferric ammonium citrate, ferrous lactate and ferrous gluconate [30], which was consistent with our recent study [31]. Hepcidin is an important iron-regulatory hormone, for which the suppression under anemic condition allows for more dietary iron to be absorbed during upregulation by oral administration of high-dose iron. This can limit the efficiency of iron absorption [32].

Interestingly, Indian school children who had been fed a high iron (12.5 mg)-fortified meal for 6 months significantly improved their physical performance, but the intervention did not influence blood Hb concentrations, biochemical parameters and cognitive function. However, a prevalence of anemia was observed to have decreased in children who were fed with a low iron (6.25 mg)-fortified meal [33]. One report has supported the contention that interventions of iron-biofortified food significantly improved the cognitive performance of subjects with regard to attention and memory domains, but did not have an effect on iron deficiency or anemia [15]. In addition, Cambodian school children who were infected with hookworm were associated with low body iron levels and potentially low cognitive performance; however, hook worm infection was found to be more prevalent in the subjects who consumed iron-fortified rice [34,35]. Moreover, consumption of regular Indian rice (1 kg) fortified with FPP (20 mg iron), folic acid (1.3 mg) and vitamin B1, B6 and B12 complexes for 8 months significantly improved the average cognitive performance scores when evaluating subjects with the Annual Status of Education Report of the Pratham Resource Center. These fortifications also increased blood Hb levels in Indian school children [36]. On the other hand, iron deficiency can result in impaired psychomotor development and cognitive function in infants and preschoolers, defective work performance in adults and low birth weights among pregnant women. With regard to neuro-protective properties, tocopherols, such as vitamin E (α-tocopherol), and tocotrienols that are major lipid-soluble chain-breaking antioxidants abundant in rice, have been reported to protect membrane integrity and lipid peroxidation, as well as to inhibit histamine 1 receptor neuronal cells [31,37,38,39]. Additionally, higher iron intakes associated with many foods have been assessed in elderly populations using dietary pattern studies based on the Reduced Rank Regression or the Food Frequency Questionnaire. It was determined that higher iron intake was associated with a high risk of type 2 diabetes, cognitive impairment, Parkinson’s disease and Alzheimer’s disease (AD) [40,41,42,43,44]. Specifically, a higher intake of dietary heme-iron, but neither non-heme-iron nor supplemental iron, could result in an increase in an accumulation of iron in the body and brain, along with a risk of iron-related disorders in the elderly. Moreover, cognitive improvement may be associated with the anti-oxidation capabilities of iron-rice drinks, as will be explained in the next section.

It has been reported that weekly consumption of micronized FPP-fortified rice (56.4 mg iron/50 g) for 18 weeks decreased the prevalence of anemia (from 31% to 19%) but did not improve blood Hb levels (before intervention: 11.4 g/dL and after intervention: 11.7 g/dL) in Brazilian infants [45]. However, polished rice that had been fortified with micronized FPP, zinc oxide, thiamin mononitrate and folic acid (50 g serving/day) was found to improve the levels of zinc, thiamine, folic acid, MCH and MCHC significantly within 4 months in Indian preschool children [46]. Likewise, twice-weekly supplementation of iron (30 mg) and folic acid (300 μg) for 20.5 weeks significantly increased blood Hb and serum Ft levels in young Cambodian children with HbA, as well as those with HbE, in which watery stool and restlessness were significantly more prevalent than in the placebo group [47]. More importantly, the treatment of Wistar rats with IR hydrolysate (50 mg/kg) for 90 days significantly increased levels of RBC, Hb, Hct and MCV when compared to the DI control group. This outcome suggests that certain active components, such as high iron content and phytochemicals in the hydrolysate, could effectively enhance erythropoiesis [48]. As is shown in Table 1, the numbers of anemic male and female subjects in group C are nearly the same, whereas there were more non-anemic female subjects in groups A, B and D than male subjects (*p* < 0.05). However, group C had the largest number of 80-year-old subjects (*p* < 0.05). Thus, increased Hb levels among the anemic elders (group G) on the first 15 days resulted from the oral intervention of the Sinlek IR rice beverage, which accelerated the duodenal absorption of dietary iron. In previous studies, women of childbearing age have been known to become anemic during pregnancy due to the massive loss of menstrual blood. In addition, around 22% of Mexican women were found to be anemic as a consequence of their level of iron deficiency (80.8% of the total population). These subjects were also reported to have a low intake of vegetables and citrus fruit that would enhance their levels of iron absorption [49]. All female subjects enrolled in this study had already undergone menopause and were not deficient in iron; therefore, gender was not a confounding factor that contributed to the increased level of hemopoietic activity in group C. During the course of aging, inflammation can have a deleterious effect on hematopoietic stem cell function, self-renewal capacity and RBC membrane integrity, all of which can lead to anemia [50,51]. Results of this study confirm that the blood hemoglobin levels of anemic elders (group C) were increased by oral administration of the iron-rice drink, but this was not the case for female subjects nor those that were of advanced age.

The consumption of FPP-premixed rice significantly increased serum ferritin levels and reduced IDA among Filipino schoolchildren when compared to non-fortified rice used as a control [52]. Furthermore, this supplemented rice diet also increased body iron stores among Indian children [46]. In a current study, healthy women revealed a fractional degree of iron absorption from iron-fortified bouillon cubes in the following order of fortification: ferrous sulfate > *Aspergillus oryzae* grown in FPP > FPP in a positively correlated manner (*p* < 0.05) [53]. Unfortunately, it has been reported that hookworm prevalence was significantly increased from 18.4% by micronutrient-fortified rice formula I (7.55 mg of iron/g) to 22.7% by micronutrient-fortified rice formula II (10.67 mg of iron/g) in Cambodian children, especially in those who live in environments with high-infection areas [35]. Notably, all elderly subjects enrolled in this study lived in the city of Chiang Mai, which is far away from the known prevalent and/or endemic areas of hookworm. Hence, the subjects were not necessarily affected by iron deficiency caused by hookworm infection. This was particularly true for the members of group C, who reported decreased SI levels when compared to the members of the other three groups. Our findings with regard to luminal absorptive iron have revealed an increase in SI and transferrin saturation at 5 h and maximal levels at 15 h in rats that consumed a single IR hydrolysate (500 mg/kg) [48]. Similar to our results, the feeding of rice containing iron-saturated lactoferrin (Lf) at doses of 100–500 mg Lf/kg BW to Wistar rats for 28 days increased RBC numbers (*p* < 0.05) and Hb values, while decreasing SI, increasing TIBC (*p* < 0.05) and unchanging Ft values in the serum when compared with normal saline feeding [54]. It is possible that the iron present in the IR drinks was used as a substrate for de novo synthesis of heme, leading to increases of Hb and RBC production and a decrease in SI. When compared with the other three groups, SI and TIBC levels in group C anemic individuals who consumed the low-dose IR drink tended to decrease during the course of this study; however, there was neither a significant decrease in TIBC (an indirect measure of Tf level) nor a decrease in SI.

Consistent with the increased serum antioxidant capacity, IR hydrolysate exerted higher antioxidant and anti-RBC hemolysis properties in vitro than the WR hydrolysate in a concentration-dependent manner [48]. This outcome was possibly due to the presence of phenolic compounds (e.g., cyanidin, ferulic acid, chlorogenic acid and so on) and γ-oryzanol [31]. In addition, treatments with Thai brown rice (var. Khao Dawk Mali 105) have revealed anti-oxidative protection against carbon tetrachloride-induced hepatotoxicity by inhibiting liver injury, lipid peroxidation, protein oxidation and DNA damage, while decreasing cytochrome P450 (CYP2E isozyme), glutathione *S*-transferase, glutathione peroxidase, superoxide dismutase and catalase activities. Furthermore, these treatments ultimately reduced the glutathione contents in the livers of laboratory rats. Notably, germinated rice was more significantly effective than un-germinated rice and white rice, respectively. This was possibly due to the existence of phenolic acids, α-oryzanol, tocotrienol and γ-amino butyric acid [54]. Moreover, antioxidant polyphenolic compounds, especially flavonoids, excluding anthocyanin, present in plant-based foods were found to exhibit certain cognitive benefits and protective effects against neurodegenerative diseases, possibly by inhibiting neuroinflammation, improving cerebrovascular blood flow and neuronal synaptic plasticity, inducing angiogenesis and neurogenesis and scavenging neurotoxins in the brain [55,56,57]. Furthermore, the consumption of polyphenolic-rich diets was found to inhibit neuro-inflammatory processes that are associated with aging people and AD patients via systemic inflammation and human gut microbiome, and may further improve their cognitive decline [58]. Hence, the major active compounds, particularly bioiron and phenolic compounds, that are present in Sinlek iron rice could effectively exert neuro- and erythropoiesis-modulating properties. Consistently, feeding black rice and brown rice (dewaxed and germinated forms) significantly lowered the plasma levels of AST and ALT activities in animals [59,60,61]. Likewise, ethyl acetate extracts of rice bran and purple rice, predominantly containing ferulic acid and quercetin, significantly decreased levels of plasma TC and LDL-C, along with hepatic lipid in mice and patients with polygenic hypercholesterolemia [62,63,64]. More importantly, the feeding of red yeast rice together with a Mediterranean diet was more effective in decreasing (*p* < 0.05) serum LDL-C levels in diabetic and dyslipidemic patients than in subjects fed with the diet alone. However, these supplemented diets did not change serum AST and ALT activities [65]. Apparently, the IR drink influenced the levels of serum lipids and liver function enzymes in elderly subjects during the course of this study, suggesting hypolipidemic and hepato-modulating effects.

Study limitations: We assessed the efficacy of the natural IR drink per se in improvement of iron deficiency and IDA in elderly subjects, but did not fortify the drinks with iron compounds or folic acid. In addition, folic acid was not added to the iron rice as a coenzyme for heme synthesis, but it may be persistent in sufficient amounts in the rice or could be derived from luminal normal flora synthesis. The intervention time was only 30 days, which is very short when compared to other clinical trials. Additionally, the responses of erythropoiesis and brain function to the iron-rich food were noticeably slower in the elderly population than in developing children and adolescents. Together, the elders who enrolled in this study did not report a reduction in iron deficiency status and may require a longer period of time to adjust their erythropoietic activity in response to the iron intervention.

## 4. Materials and Methods

### 4.1. Materials

#### 4.1.1. Chemicals and Reagents

6-Hydroxy−2,5,7,8-tetramethylchroman−2-carboxylic acid (Trolox) was purchased from Sigma-Aldrich Chemicals, St. Louis, MO, USA. Reagents and calibrators for the analyses of serum cholesterol, triglyceride, glucose, protein and albumin were purchased from Randox Laboratories, Crumin, United Kingdom. Reagents and calibrators for the analysis of SI, TIBC and serum Ft were purchased from Roche Diagnostics Corporation, Indianapolis, IN, United States. All other materials and solvents were of the highest purity or HPLC grade.

#### 4.1.2. Food Ingredients

Sinlek iron rice and Jasmin white rice were purchased from a private organic rice farm located in Doi Saked District, Chiang Mai Province, Thailand. Citric acid monohydrate (food additive, Special Food Supplier, Samuthpakarn, Thailand) without detectable levels of iron, mercury and lead was purchased from Tesco Supermarket in Chiang Mai Province, Thailand. Polyflora bee honey (Food-grade, Eurngluang Brand) was obtained from Polyplus Company Limited, Muang Chiang Mai, Thailand, for which 100 g contained 1.12 g protein, 0 g fats, 8.89 g carbohydrates, 72.28 g natural sugars, 0.75 g dietary fiber, 1.06 mg sodium, 0.33 mg vitamin C, 4.3 mg calcium and 0.21 mg iron. A heat-stable α-amylase (Termamyl^®^, Novozymes™) obtained from *Bacillus* spp. was purchased from Sigma-Aldrich Chemicals, St. Louis, MO, USA.

### 4.2. Production of Rice Drink

The rice drink was produced, and chemical compositions were analyzed using the recently published method established by Koonyosying and colleagues [31]. Briefly, rice grains were polished, ground, filtered on nylon 300-μm mesh fabric (Hebei Shangshai Bolting Cloth Manufacturing Company Limited, Shandon, People’ Republic of China) and boiled in hot water (6 kg/60 L) at 80 °C for 10 min. After cooling, the rice slurry was incubated with the α-amylase (Termamyl^®^) at 80 °C for 120 min and digestion was stopped by the immediate heating of the hydrolysate at 100 °C for 10 min. We then used the Design Expert version 6.2.10 Program, Product- and Consumer-Oriented tests, to optimize the ingredient composition and obtain the most suitable products. Rice hydrolysate was filtered, concentrated to 20-Brig dryness, using a freeze-dry lyophilizing machine, acidified to a pH value of 4.0 with 1% (*v*/*v*) citric acid, flavored with bee honey, poured into 100 mL glass bottles and sterilized at standard temperature and pressure values. From the inductively coupled plasma mass spectrometric analysis, WR (2 mg iron/100 mL serving), low-dose IR (15 mg iron/100 mL serving) and high-dose IR (27 mg iron/100 mL serving) were provided to each group of the subjects throughout the course of the study [66]. In addition, carbohydrates, protein, fat and the bioactive compounds, including total phenolic (1.15, 1.33 and 2.08 mg gallic acid equivalent, respectively), γ-oryzanol (54, 193 and 316 mg, respectively) and antioxidant capacity (111, 856 and 1400 mg Trolox equivalent, respectively), were comprised in the drinks [31]. Using liquid chromatographic/mass spectrometric analysis, the IR drink has shown the presence of several phenolic compounds, *p*-protocatechuoyl-*O*-glucoside and kaempherol, which were not found in the WR drink [31]. Vitamin C or L-ascorbic acid was analyzed using the high-performance liquid chromatography-diode array detection (HPLC-DAD) method established by Lakshanasomya [67]. Briefly, WR and IR drinks and standard vitamin C (2.5–30 μg/mL) were prepared in 3% meta-phosphoric acid, and the samples (20 μL) were subsequently injected into the HPLC-DAD system (Agilent Series 1100, Agilent Technologies Inc., Santa Clara, CA, USA) connected to a reverse-phase column (C18 type, 125 × 4 mm, 5 μm particle size, Lichrocard, Lichrospher 100, Merck Millipore, Merck KGaA, Darmstadt, Germany). Both were then isocratic eluted with the mobile-phase solvent (3 mM phosphate buffer, pH 3.3 in 0.35% (*v*/*v*) 3% meta-phosphoric acid at a flow rate of 0.5 mL/min), while the OD of the eluents was detected at 248 nm. The vitamin C peak of the rice drinks was compared with that of authentic vitamin C at the same retention time and used for the quantification standard established from the calibration curve. In the analysis, the WR drink contained 0.04 mg vitamin C/100 mL per serving, while the low-dose and high-dose IR drinks contained 0.13 and 0.38 mg vitamin C/100 mL per serving, respectively. Essentially, 3.7% (*v*/*v*) bee honey as a flavoring agent and 1% citric acid as a stabilizing agent were added to the drinks.

### 4.3. Subject Recruitment

According to the inclusion criteria, elderly subjects enrolled in this study were aged >60 years living in Chiang Mai and were able to communicate in Thai. The exclusion criteria included the following: ADL score ≤ 11, history of thalassemia, uncontrolled diabetes with FBS > 130 mg/dL or postprandial blood sugar > 180 mg/dL, abnormal bleeding within 3 m, cancer of colon, esophagus, stomach, small intestine, or ampullary, angiodysplasia, celiac disease, gastric antral vascular ectasia, blood donation less than 3 months, *Helicobacter pylori*-infected gastritis, esophagitis, gastrectomy/colectomy, or a period of treatment with proton-pump inhibitors, iron formulations or folic acid.

The sample size (n) was calculated using the following Formula (1):N = (Z_1_ − α/2 + Z_1_ − β)^2^ [σ^2^_treatment_ + σ^2^_control_/r]/∆^2^(1)
where r = n_control_/n_treatment_, Δ = µ_treatment_ − µ_control_, α = 0.01, β = 0.2, µ_treatment_ = 11, µ_control_ = 10 and SD = 1. According to this calculation, 25 volunteers were placed in each group and a value of 10% was then added as a calculated number for the purpose of loss during follow-up. Ultimately, 27 subjects in each group were recruited for this study. We recruited healthy subjects from elderly schools or elderly clubs in Muang Chiang Mai. The elderly subjects were interviewed in order to establish their demographic information. They were also subjected to a psychological test (cognition and emotion), nutritional evaluation, activities and daily living assessment, a physical examination and a blood test screening. The elderly subjects who participated in the study were asked to sign documents of informed consent. Subsequently, we matched 3 non-anemic elderly subjects by sex, age and Hb level and 1 anemic elderly subject by sex and age in each elderly school and/or elderly club.

### 4.4. Administration of Rice Drinks

Elderly volunteers who were recruited according to the inclusion criteria were divided into four groups. There were three groups of non-anemic (*n* = 77) subjects and one group of anemic (*n* = 27) subjects. The non-anemic volunteers in each group were randomly allocated to receive the WR drink (group A, *n* = 27), the low-dose IR drink (group B, *n* = 27) and the high-dose IR drink (group D, *n* = 27) (one 100 mL bottle/day), while the anemic group (group C, *n* = 27) received only the low-dose IR drink (one 100 mL bottle/day) for 1 month. All subjects were then asked to fast and to only drink water for 12 h. Subsequently, blood samples were collected from veins of all subjects on days 0, 15 (±1) and 30 (±1 or 2).

### 4.5. Physical Examination, Psychological and Nutrition Testing

BW, SBP/DBP and pulse rate were recorded on days 0 and 30. The MMSE Thai 2002 and PHQ9 tests (Thai-version) were used to assess whether there were any issues of cognitive ability and/or depression on days 0 and 30 (±1 or 2), respectively [68,69,70]. Next, a questionnaire on the nutritional background of each subject was used to complete a nutritional assessment. Lastly, Barthel’s index was used to evaluate the activities of daily life for each subject.

### 4.6. Blood Analysis

For the hematological parameters, a CBC was measured using an automated Cell Analyzer (Beckman-Coulter Inc., Brea, CA. USA) according to the manufacturer’s instructions. Biochemical parameters were measured using an automated Clin Chem Analyzer (Randox^®^, Randox Laboratories Company Limited, County Antrim, UK) according to the manufacturer’s instructions. SI and TIBC were measured using an automated ferrozine colorimetric analyzer (Cobas, Roche Diagnostics Corporation, Indiana, IN, USA) and serum Ft was quantified using an Elecsys^®^ Chemiluminescent immunoassay analyzer (Cobas, Roche Diagnostics Corporation, Indianapolis, IN, USA) according to the manufacturer’s instructions [71,72].

### 4.7. Statistical Analysis

Data are presented in percentages and values of frequency, mean ± SD and mean ± SEM. Statistical significance was analyzed using Stata 16.0 software, for which *p* < 0.05 was considered a significant difference. Volunteer characteristics and health information of the members of all four groups were analyzed by the Chi-square test. Before and after variables were compared by the paired Student’s t-test for parametric data and Wilcoxon signed rank test for non-parametric data. Biochemistry variables were analyzed by the Shapiro–Wilk test to establish a degree of normality. A comparison of variable mean values in 1, 2 and 3 visits for each group was achieved using the repeated ANOVA test for normal distribution variables and paired Student’s t-test for paired visits. Mean values of non-normal distribution variables in 1, 2 and 3 visits were analyzed using the Friedman test and Wilcoxon signed rank test for paired visits. Normal distribution mean changes of all groups for the same visit were compared by ANOVA, and these values were compared between groups using the Bonferroni test. Furthermore, non-distribution mean change values were analyzed by the Kruskal–Wallis test and comparisons between groups were made by the Dunn test.

## 5. Conclusions

This is the first functional drink that was produced from non-iron-fortified, non-transgenic rice, and it comprised higher iron content and antioxidant capacity than the white-rice drink. Importantly, consumption of the iron-rice drink for at least 15 days functioned to increase hemoglobin and antioxidant capacity levels in anemic elderly subjects. However, long-term consumption of the iron-rice drink is recommended to improve iron-deficiency anemia in aging people. Furthermore, folic acid fortification of iron rice may be necessary to promote iron-enhanced erythropoiesis.

## Figures and Tables

**Figure 1 molecules-26-06285-f001:**
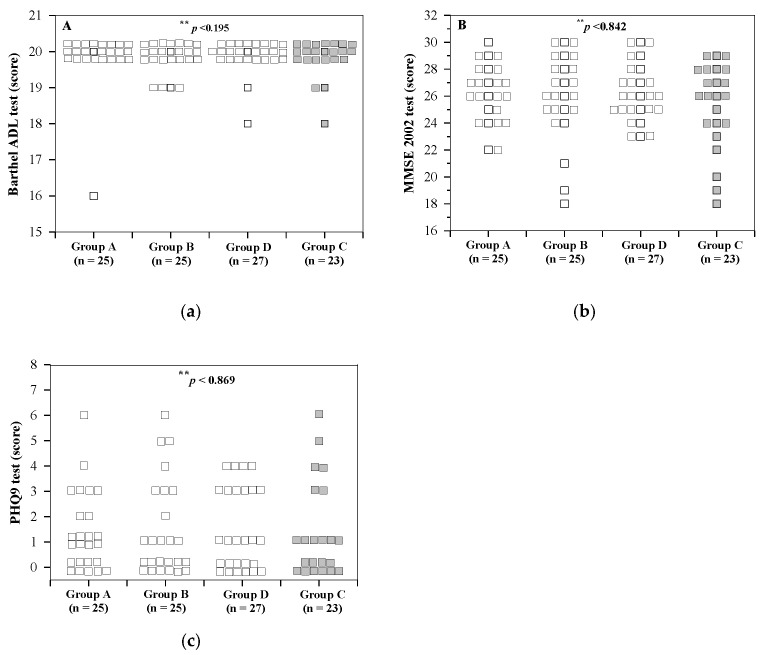
Numbers of subjects in groups A, B, D and C for Barthel ADL (**a**), MMSE Thai 2002 (**b**) and PHQ9 (**c**) tests before the study. Non-anemic elders consumed the WR drink (2 mg iron/100 mL serving) (group A), IR drink (15 mg iron/100 mL serving) (group B) and IR drink (27 mg iron/100 mL serving) (group D), while anemic elders consumed the IR drink (15 mg iron/100 mL serving) (Group C) for 30 days. Data of individual subjects are expressed accordingly. ** Fisher’s exact test when members of the same group were compared. Abbreviations: ADL = activities of daily living, MMSE = Mini Mental State Examination, PHQ9 = Patient Health Questionnaire 9.

**Figure 2 molecules-26-06285-f002:**
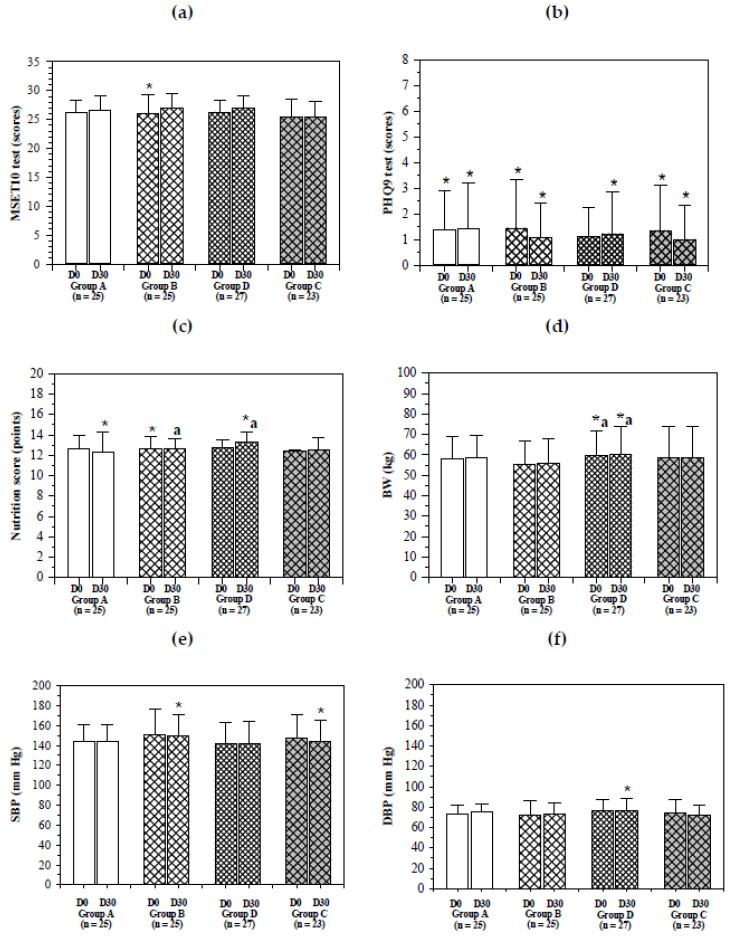
Changes in neurological and nutritional scores, body weight and blood pressure values in elderly subjects. Non-anemic elders consumed the WR drink (2 mg iron/100 mL serving) (group A), IR drink (15 mg iron/100 mL serving) (group B) and the IR drink (27 mg iron/100 mL serving) (group D), while anemic elders (group C) consumed the IR drink (15 mg iron/100 mL serving) for 30 days and reported MMSE Thai 2002 (**a**), PHQ9 (**b**) and nutritional (**c**) scores, BW (**d**) and BP (**e**,**f**) values. Data are expressed as mean ± standard deviation (SD) values. * Shapiro–Wilk W test (*p* < 0.05). ^a^ Wilcoxon signed rank-sum test *p* < 0.05 when members of the same group were compared before and after the study. Abbreviations: BW = body weight, BP = blood pressure, DBP = diastolic blood pressure, Hg = mercury, IR = iron rice, MMSE Thai 2002 = Mini Mental State Examination Thai 2002, PHQ9 = Patient Health Questionnaire 9, SBP = systolic blood pressure, WR = white rice.

**Figure 3 molecules-26-06285-f003:**
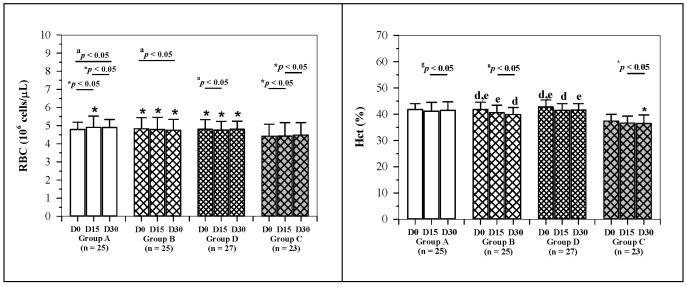
Changes in RBC indices, WBC and platelet numbers of elderly subjects. Non-anemic subjects consumed the WR drink (2 mg iron/100 mL serving) (group A), IR drink (15 mg iron/100 mL serving) (group B) and IR drink (27 mg iron/100 mL serving) (groups D), and anemic subjects consumed the IR drink (15 mg iron/serving) (group C) for 30 days and blood samples were collected for complete blood count analysis. Data are expressed as mean ± SD values. * Comparisons of results of Shapiro–Wilk test (*p* < 0.05). ^d,e,f^ Between D0 and D15, D0 and D30, D15 and D30 for normal distribution by paired Student’s *t*-test (*p* < 0.05) after using repeated analysis of variance (repeated ANOVA) (*p* < 0.05). ^a,b,c^ Comparisons between D0 and D15, D0 and D30 and D15 and D30 for non-normal distribution were performed by the Wilcoxon signed rank test (*p* < 0.05) after using the Friedman test (*p* < 0.05). ^j,k,l^ Bonferroni test (*p* < 0.05) among groups with normal distribution data after using one-way analysis of variance (one-way ANOVA) test. ^g,h,i^ Comparisons of Kruskal–Wallis test (*p* < 0.05) for four groups and Dunn test (*p* < 0.05) between two groups with non-normal distribution data. Abbreviations: EOS = eosinophil, Hb = hemoglobin, Hct = hematocrit, Lym = lymphocyte, MCH = mean corpuscular hemoglobin, MCHC = mean corpuscular hemoglobin, MCV = mean corpuscular volume, Mon = monocytes, PLT = platelet, PMN = polymorphonuclear cell, RBC = red blood cells, WBC = white blood cells.

**Figure 4 molecules-26-06285-f004:**
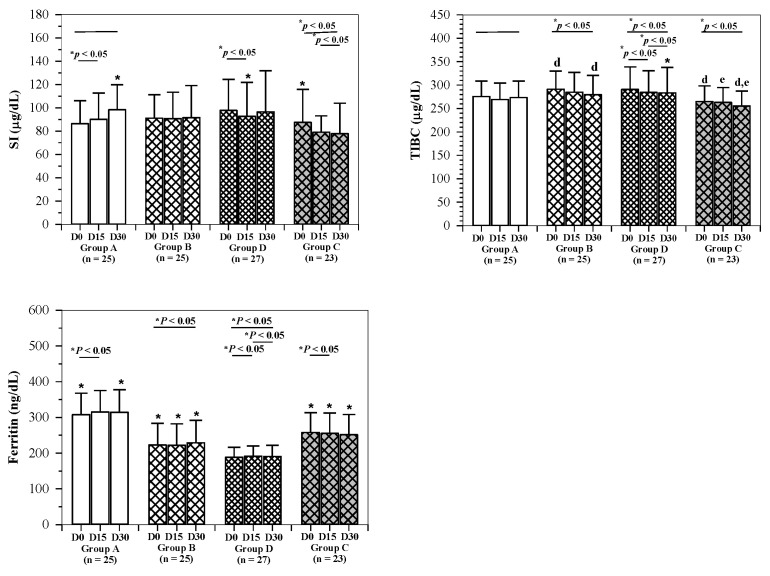
Changes in levels of serum iron, total iron-binding capacity and ferritin of elderly subjects. Non-anemic elders consumed the WR drink (2 mg iron/100 mL serving) (group A), IR drink (15 mg iron/serving) (group B) and IR drink (27 mg iron/100 mL serving) (group D), and anemic elders consumed the IR drink (15 mg iron/100 mL serving) (group C) for 30 days and blood samples were collected for analysis of iron parameters. Data are expressed as mean ± SD values with the exception of ferritin, which is expressed as mean ± SEM values. * Shapiro–Wilk test (*p* < 0.05). ^d,e,f^ Comparisons between D0 and D15, D0 and D30, D15 and D30 for normal distribution by paired Student’s t-test (*p* < 0.05) after using repeated measure ANOVA (*p* < 0.05). Abbreviations: Ft = ferritin, SI = serum iron, TIBC = total iron-binding capacity.

**Figure 5 molecules-26-06285-f005:**
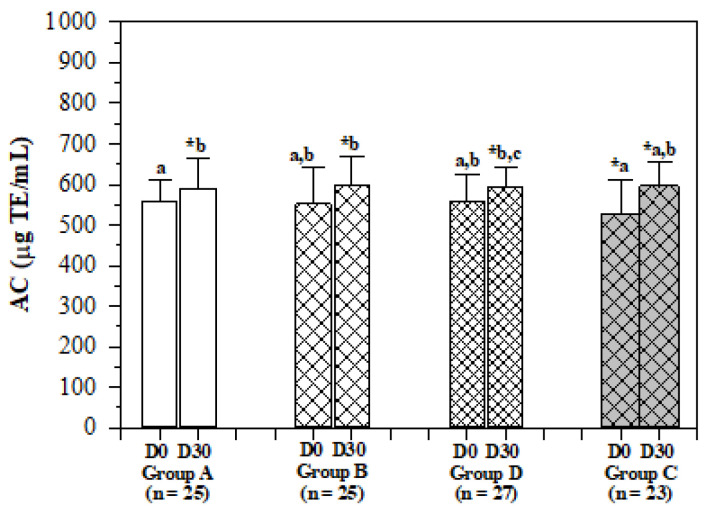
Changes in levels of serum antioxidant capacity of elderly subjects. Non-anemic elders consumed the WR drink (2 mg iron/100 mL serving) (group A), IR drink (15 mg iron/100 mL serving) (group B) and IR drink (27 mg iron/100 mL serving) (group D), while anemic elders consumed the IR drink (15 mg iron/100 mL serving) (group C) for 30 days and blood samples were collected for analysis of antioxidant capacity. Data are expressed as mean ± SD values with the exception of ferritin, which is expressed as mean + standard error of the mean (SEM). * Shapiro–Wilk test (*p* < 0.05). ^a,b,c^ Comparisons between D0 and D30 for non-normal distribution by Wilcoxon signed rank test (*p* < 0.05) after applying the Friedman test (*p* < 0.05). Abbreviations: AC = antioxidant capacity, TE = Trolox equivalent.

**Table 1 molecules-26-06285-t001:** General information of subjects in groups A, B, D and C before the study. Non-anemic elders consumed a WR drink (2 mg iron/100 mL serving) (group A), IR drink (15 mg iron/100 mL serving) (group B) and IR drink (27 mg iron/100 mL serving) (group D), while anemic elders consumed the IR drink (15 mg iron/100 mL serving) (Group C). Data are expressed in absolute numbers and percentages (blanket). * Pearson chi-square test, ** Fisher’s exact test (*p* < 0.05) when members of the same group were compared.

Characteristics		Number (Percentage) of Subjects				*p*-Value When All Groups Were Compared
		Non-Anemic			Anemic	
		Group A	Group B	Group D	Group C	
Gender	Male	7 (28.00)	4 (16.00)	2 (7.41)	12 (52.17)	*p* = 0.002 *
	Female	18 (72.00)	21 (84.00)	25 (92.59)	11 (47.83)	
Age (years)	60–69	13 (52.00)	10 (40.00)	18 (66.67)	5 (21.74)	*p* = 0.019 **
	70–79	11 (44.00)	11 (44.00)	5 (18.52)	12 (52.17)	
	≥80	1 (4.00)	4 (16.00)	4 (14.81)	6 (26.09)	
Marital status	Single	3 (12.00)	3 (12.00)	3 (11.11)	3 (13.04)	*p* = 0.491 **
	Couple	13 (52.00)	11 (44.00)	10 (37.04)	8 (34.78)	
	Divorce/separate	0	0	3 (11.11)	4 (17.39)	
	Widow	9 (36.00)	11 (44.00)	11 (40.74)	8 (34.78)	
Education	Primary school	15 (60.00)	18 (72.00)	17 (62.96)	13 (56.62)	*p* = 0.485 **
	Secondary school	0	1 (4.00)	0	2 (8.70)	
	High school	6 (24.00)	0	5 (18.52)	3 (13.04)	
	Under graduate	2 (8.00)	3 (12.00)	1 (3.70)	2 (8.70)	
	Postgraduate	0	1 (4.00)	1 (3.70)	1 (4.35)	
	Other	2 (8.00)	2 (8.00)	3 (11.11)	2 (8.70)	
Profession	No	16 (64.00)	13 (52.00)	18 (66.67)	10 (43.48)	*p* = 0.704 **
	Agriculture	0	1 (4.00)	0	3 (13.04)	
	Worker	0	2 (8.00)	1 (3.70)	3 (13.04)	
	Merchant	3 (12.00)	3 (12.00)	2 (7.41)	2 (8.70)	
	Pension	2 (8.00)	2 (8.00)	3 (11.11)	1 (4.35)	
	Other	4 (16.00)	4 (16.00)	3 (11.11)	4 (17.39)	

**Table 2 molecules-26-06285-t002:** Biographic information of subjects in groups A, B, D and C before the study. Non-anemic elders consumed a WR drink (2 mg iron/100 mL serving) (group A), IR drink (15 mg iron/100 mL serving) (group B) and IR drink (27 mg iron/100 mL serving) (group D), while anemic elders consumed the IR drink (15 mg iron/100 mL serving) (Group C). Data are expressed in absolute numbers and percentages (blanket). * Pearson chi-square test, ** Fisher’s exact test (*p* < 0.05) when members of the same group were compared.

Health Information		Number (Percentage) of Subjects				*p*-Value When All Groups Were Compared
		Non-Anemic			Anemic	
		Group A	Group B	Group D	Group C	
BMI (kg/m^2^)	<18.5	0	1 (4.00)	0	1 (4.35)	*p* = 0.911 **
	18.5–22.9	9 (36.00)	10 (40.00)	10 (37.04)	7 (30.43)	
	≥23	16 (64.00)	14 (56.00)	17 (62.96)	15 (65.22)	
Chronic diseases		17/25 (68.00)	12/25 (48.00)	18/27 (66.67)	19/23 (82.91)	*p* = 0.090 *
Hypertension		11/25 (44.00)	9/25 (36.00)	13/27 (48.15)	12/23 (52.17)	*p* = 0.702 *
Diabetic mellitus		0/25 (0)	3/25 (12.00)	3/27 (11.11)	3/23 (13.04)	*p* = 0.287 **
Hyperlipidemia		9/25 (36.00)	7/25 (28.00)	12/27 (44.44)	8/23 (34.78)	*p* = 0.672 *
Osteoarthritis		5/25 (20.00)	4/25 (16.00)	4/27 (14.81)	7/23 (30.43)	*p* = 0.541 **
Smoking		1/25 (4.00)	0/25 (0)	1/27 (3.70)	2/23 (3.70)	*p* = 0.456 **
Alcohol drinking		1/25 (4.00)	0/25 (0)	0/27 (0)	2/23 (8.70)	*p* = 0.136 **
Exercise		21/25 (84.00)	19/25 (76.00)	20/27 (74.07)	16/23 (69.57)	*p* = 0.693 *
Drug allergy		4/25 (16.00)	0/25 (0)	5/27 (18.52)	1/23 (4.35)	*p* = 0.063 **

Abbreviations: BMI = body mass index.

**Table 3 molecules-26-06285-t003:** Levels of serum glucose and lipid profiles of elderly subjects. Non-anemic elders consumed the WR drink (2 mg iron/100 mL serving) (group A), IR drink (15 mg iron/100 mL serving) (group B) and IR drink (27 mg iron/100 mL serving) (group D), and anemic elders consumed the IR drink (15 mg iron/100 mL serving) (group C) for 30 days and blood samples were collected for analysis of glucose and lipids. Data are expressed as mean ± SD values. Changes in the mean are shown as mean ± SD values in the brackets.

Parameters	Time	Group A (*n* = 25)	Group B (*n* = 25)	Group D (*n* = 27)	Group C (*n* = 23)
FBS (mg/dL)	D0 (D0:D15)	82.5 ± 13.2 (7.3 ± 3.4)	79.0 ± 14.6 ^d,e^ (6.5 ± 2.5)	81.0 ± 14.5 ^d^ (7.3 ± 3.1 *)	84.4 ± 16.1 (1.8 ± 3.2)
	D15 (D15:D30)	88.8 ± 15.6 (4.5 ± 5.0 *)	85.5 ± 17.7 ^d^ (−1.4 ± 1.8 *)	87.9 ± 14.0 ^d^ (−3.8 ± 2.9 *)	86.2 ± 16.4 (6.3 ± 3.3 *)
	D30 (D0:D30)	93.2 ± 26.1 * (11.8 ± 6.1 *)	84.1 ± 15.6 ^e^ (5.1 ± 2.1)	85.3 ± 15.2 (3.5 ± 2.9)	92.5 ± 21.9 (8.1 ± 4.6 *)
TC (mg/dL)	D0 (D0:D15)	210 ± 39 (−11 ± 4 *)	209 ± 37 ^d^ (−8 ± 5)	207 ± 36 (−1 ± 5)	193 ± 30 (−4 ± 4)
	D15 (D15:D30)	203 ± 38 (7 ± 5)	202 ± 32 (−4 ± 4)	206 ± 39 (1 ± 3)	188 ± 32 (4 ± 4)
	D30 (D0:D30)	209 ± 37 (5 ± 12 *)	198 ± 29 ^d^ (0 ± 8)	207 ± 39 (11 ± 7)	192 ± 35 (9 ± 13 *)
TG (mg/dL)	D0 (D0:D15)	120 ± 59 (−22 ± 13 *)	117 ± 54 * (−6 ± 11)	105 ± 36 (10 ± 10)	110 ± 54 (5 ± 8)
	D15 (D15:D30)	98 ± 36 ^a^ (27 ± 9)	111 ± 62 * (6 ± 12 *)	113 ± 50 * (1 ± 7)	115 ± 52 (4 ± 9)
	D30 (D0:D30)	125 ± 48 *^a^ (5 ± 12 *)	117 ± 71 * (0 ± 8)	114 ± 37 (11 ± 7)	119 ± 61 * (9 ± 12 *)
HDL-C (mg/dL)	D0 (D0:D15)	53 ± 11 (−5 ± 2 *)	54 ± 17 *^a^ (−3 ± 1)	55 ± 12 ^d^ (−3 ± 1 *^)^	51 ± 14 ^d^ (−5 ± 1)
	D15 (D15:D30)	49 ± 13 *^a^ (7 ± 2 *)	51 ± 15 *^a,b^ (4 ± 2)	53 ± 12 ^d^ (3 ± 2 *)	46 ± 13 ^d,e^ (4 ± 1)
	D30 (D0:D30)	55 ± 12 ^a^ (2 ± 2)	56 ± 17 *^b^ (1 ± 1 *)	56 ± 15 (0 ± 2)	49 ± 14 ^e^ (−1 ± 1)
LDL-C (mg/dL)	D0 (D0:D15)	169 ± 45 (−20 ± 7 *)	159 ± 45 *^a,b^ (−12 ± 5)	155 ± 41 (−7 ± 5)	146 ± 36 ^d^ (−5 ± 1)
	D15 (D15:D30)	153 ± 48 (5 ± 6 *)	147 ± 37 ^a^ (−4 ± 4 *)	148 ± 48 *^a^ (7 ± 4)	134 ± 35 ^d,e^ (10 ± 4)
	D30 (D0:D30)	157 ± 45 (−16 ± 8)	143 ± 33 ^b^ (−16 ± 6)	155 ± 46 *^a^ (−1 ± 4)	144 ± 42 ^e^ (−1 ± 6)

* Comparisons between results for the Shapiro–Wilk test (*p* < 0.05). ^d,e,f^ Comparison of D0 and D15, D0 and D30 and D15 and D30 for normal distribution by paired Student’s *t*-test (*p* < 0.05) after using repeated ANOVA (*p* < 0.05). ^a,b,c^ Comparisons between D0 and D15, D0 and D30 and D15 and D30 for non-normal distribution by Wilcoxon signed rank test (*p* < 0.05) after using the Friedman test (*p* < 0.05). Abbreviations: FBS = fasting blood sugar, HDL-C = high-density lipoprotein-cholesterol, LDL-C = low-density lipoprotein-cholesterol, TC = total cholesterol, TG = triglyceride.

**Table 4 molecules-26-06285-t004:** Levels of blood urea nitrogen, serum creatinine, uric acid and electrolytes of non-anemic subjects who consumed the WR drink (2 mg iron/100 mL serving) (group A), IR drink (15 mg iron/100 mL serving) (group B) and IR drink (27 mg iron/100 mL serving) (group D), and anemic subjects who consumed the IR drink (15 mg iron/100 mL serving) (group C) for 30 days. Data are expressed as mean ± SD values. Changes in the mean are shown as mean ± SD values in the brackets.

Parameters	Time	Group A (*n* = 25)	Group B (*n* = 25)	Group D (*n* = 27)	Group C (*n* = 23)
BUN (mg/dL)	D0 (D0:D15)	13.2 ± 2.4 ^d^ (−1.7 ± 0.4)	13.0 ± 3.5 (−0.9 ± 0.4)	14.0 ± 3.8 (−1.1 ± 0.6)	15.8 ± 6.9 * (−0.7 ± 0.8)
	D15 (D15:D30)	11.6 ± 2.1 ^d^ (0.5 ± 0.4)	12.1 ± 2.9 * (0.8 ± 0.6)	12.8 ± 2.8 (0.4 ± 0.6)	15.1 ± 6.6 * (0.2 ± 0.6 *)
	D30 (D0:D30)	12.2 ± 2.2 (−1.2 ± 0.6 *)	12.9 ± 2.8 (−0.1 ± 0.7)	13.0 ± 2.9 * (−0.7 ± 0.7)	15.3 ± 5.2 * (−0.4 ± 1.0 *)
CRE (mg/dL)	D0 (D0:D15)	0.90 ± 0.14 (0.01 ± 0.03)	0.87 ± 0.20 * (0.03 ± 0.03)	0.87 ± 0.19 * (0 ± 0.04)	1.21 ± 0.61 * (0.01 ± 0.04)
	D15 (D15:D30)	0.91 ± 0.16 (0 ± 0.04)	0.90 ± 0.19 (0.05 ± 0.03)	0.87 ± 0.17 * (0.03 ± 0.04)	1.22 ± 0.62 * (0.03 ± 0.04)
	D30 (D0:D30)	0.94 ± 0.29 (0.01 ± 0.06 *)	0.95 ± 0.21 (0.08 ± 0.04 *)	0.90 ± 0.16 (0.03 ± 0.05)	1.25 ± 0.70 * (0.04 ± 0.05)
UA (mg/dL)	D0 (D0:D15)	6.75 ± 2.07 * (0.07 ± 0.41 *)	7.35 ± 2.10 (−0.23 ± 0.21)	6.24 ± 1.86 * (0.10 ± 0.46 *)	7.63 ± 2.56 (0.43 ± 0.55 *)
	D15 (D15:D30)	6.85 ± 1.90 (0.17 ± 0.41)	7.12 ± 1.71 * (0.31 ± 0.21)	6.31 ± 1.36 (0.16 ± 0.27)	8.06 ± 2.57 (−0.16 ± 0.37 *)
	D30 (D0:D30)	7.03 ± 2.03 (0.25 ± 0.50 *)	7.43 ± 1.69 (0.08 ± 0.25)	6.43 ± 1.26 (0.26 ± 0.42 *)	7.90 ± 1.75 (0.27 ± 0.45 *)
Na^+^ (mmol/L)	D0 (D0:D15)	142 ± 3 ^d,e^ (−1 ± 1)	144 ± 3 *^a,b^ (−3 ± 1)	143 ± 2 (−2 ± 1 ^a^)	140 ± 3 (−1 ± 1 ^a^)
	D15 (D15:D30)	140 ± 2 ^d^ (−0 ± 1)	141 ± 3 *^a,c^ (−1 ± 1)	141 ± 2 (−2 ± 1)	139 ± 2 (−0 ± 1)
	D30 (D0:D30)	140 ± 3 ^e^ (−2 ± 1)	139 ± 3 ^b,c^ (−4 ± 1)	140 ± 2 (−3 ± 1 ^a^)	139 ± 3 (−1 ± 1 ^a^)
K^+^ (mmol/L)	D0 (D0:D15)	3.80 ± 0.44 (0 ± 0.08)	4.03 ± 0.35 ^d,e^ (−0.12 ± 0.05)	4.09 ± 0.80 * (−0.24 ± 0.18 *)	4.27 ± 0.51 ^a^ (−0.17 ± 0.19 *)
	D15 (D15:D30)	3.77 ± 0.51 (−0.05 ± 0.80)	3.91 ± 0.36 ^e^ (−0.1 ± 0.05)	3.92 ± 0.42 * (0.01 ± 0.11)	4.03 ± 0.75 *^a,b^ (0.05 ± 0.14 *)
	D30 (D0:D30)	3.74 ± 0.44 (−0.04 ± 0.08)	3.81 ± 0.31 ^d^ (−0.22 ± 0.06)	3.95 ± 0.57 * (−0.15 ± 0.20 *)	4.08 ± 0.46 ^b^ (−0.19 ± 0.10)

* Comparisons of results of the Shapiro–Wilk test (*p* < 0.05). ^d,e,f^ Between D0 and D15, D0 and D30 and D15 and D30 for normal distribution by paired Student’s t-test (*p* < 0.05) after using repeated ANOVA (*p* < 0.05). ^a,b,c^ Comparisons between D0 and D15, D0 and D30 and D15 and D30 for non-normal distribution by Wilcoxon signed rank test (*p* < 0.05) after using the Friedman test (*p* < 0.05). Abbreviations: BUN = blood urea nitrogen, CRE = creatinine, K^+^ = potassium ion, Na^+^ = sodium ion, UA = uric acid.

**Table 5 molecules-26-06285-t005:** Levels of serum proteins, aspartate aminotransferase and alanine aminotransferase activity of non-anemic subjects who consumed the WR drink (2 mg iron/100 mL serving) (group A), IR drink (15 mg iron/100 mL serving) (group B) and IR drink (27 mg iron/100 mL serving) (group D), while anemic subjects consumed the IR drink (15 mg iron/100 mL serving) (group C) for 30 days. Data are expressed as mean ± SD values. Changes in the mean are shown as mean ± SD values in the brackets.

Parameters	Time	Group A (*n* = 25)	Group B (*n* = 25)	Group D (*n* = 27)	Group C (*n* = 23)
TP (g/dL)	D0 (D0:D15)	8.00 ± 0.46 (−0.24 ± 0.09)	8.14 ± 0.72 ^d,e^ (−0.11 ± 0.15)	7.97 ± 0.40 (−0.00 ± 0.12)	7.94 ± 0.35 ^d^ (−0.24 ± 0.09)
	D15 (D15:D30)	7.96 ± 0.42 * (−0.09 ± 0.07)	7.90 ± 0.56 ^e^ (1.99 ± 0.09)	7.90 ± 0.54 (−0.34 ± 0.12)	7.94 ± 0.48 ^e^ (−0.09 ± 0.07)
	D30 (D0:D30)	7.82 ± 0.46 (−0.33 ± 0.10)	7.81 ± 0.52 ^d^ (−0.11 ± 0.14 *)	7.83 ± 0.53 * (−0.34 ± 0.09)	7.60 ± 0.47 ^d,e^ (−0.33 ± 0.10)
Alb (g/dL)	D0 (D0:D15)	4.50 ± 0.36 (−0.15 ± 0.08)	4.55 ± 0.32 * (−0.09 ± 0.10)	4.44 ± 0.43 ^d^ (0.00 ± 0.12 *)	4.20 ± 0.64 ^d^ (−0.15 ± 0.08)
	D15 (D15:D30)	4.46 ± 0.51 (0.21 ± 0.05)	4.40 ± 0.43 *^a^ (0.37 ± 0.10)	4.36 ± 0.52 ^e^ (0.51 ± 0.14)	4.20 ± 0.60 ^e^ (0.21 ± 0.05)
	D30 (D0:D30)	4.66 ± 0.37 (0.06 ± 0.08)	4.62 ± 0.42 *^a^ (0.27 ± 0.09)	4.68 ± 0.38 ^d,e^ (0.51 ± 0.17 *)	4.71 ± 0.48 ^d,e^ (0.06 ± 0.08)
Glo(g/dL)	D0 (D0:D15)	3.56 ± 0.72 ^e^ (−0.05 ± 0.10)	3.57 ± 0.77 ^e^ (0.00 ± 0.16 *)	3.53 ± 0.59 *^a^ (−0.03 ± 0.14)	3.74 ± 0.75 ^d^ (−0.05 ± 0.10)
	D15 (D15:D30)	3.48 ± 0.63 ^d^ (−0.32 ± 0.09 ^h^)	3.52 ± 0.80 ^d^ (−0.39 ± 0.13 ^i^)	3.57 ± 0.64 ^b^ (−0.83 ± 0.17 ^g,h,i^)	3.72 ± 0.84 ^e^ (−0.32 ± 0.09 ^h^)
	D30 (D0:D30)	3.16 ± 0.56 ^d,e^ (−0.38 ± 0.09)	3.20 ± 0.71 ^d,e^ (−0.39 ± 0.15)	3.15 ± 0.63 *^a,b^ (−0.86 ± 0.16)	2.89 ± 0.61 ^d,e^ (−0.38 ± 0.09)
AST (U/L)	D0 (D0:D15)	19 ± 6 (−1 ± 1)	19 ± 4 (−1 ± 2 *)	20 ± 10 * (−1 ± 1)	18 ± 4 (−1 ± 1)
	D15 (D15:D30)	18 ± 4 (0 ± 1)	18 ± 3 (−1 ± 1)	20 ± 8 (−0 ± 1)	18 ± 4 (0 ± 1)
	D30 (D0:D30)	18 ± 5 (−1 ± 1)	18 ± 4 (−2 ± 2 *)	19 ± 7 * (−1 ± 1)	17 ± 4 (−1 ± 1)
ALT (U/L)	D0 (D0:D15)	19 ± 9 *^a,b^ (−4 ± 1)	20 ± 5 ^d,e^ (−4 ± 1)	21 ± 12 *^a,b^ (−4 ± 1 *)	19 ± 8 ^d,e^ (−4 ± 1)
	D15 (D15:D30)	15 ± 5 ^a^ (−0 ± 1)	15 ± 3 ^e^ (−0 ± 1)	17 ± 9 *^a^ (−1 ± 1)	15 ± 6 ^d^ (−0 ± 1)
	D30 (D0:D30)	15 ± 4 ^b^ (−5 ± 1)	15 ± 5 ^d^ (−4 ± 1)	17 ± 10 *^b^ (−5 ± 1 *)	14 ± 5 ^e^ (−5 ± 1)

* Comparisons of results of the Shapiro–Wilk test (*p* < 0.05). ^d,e,f^ Between D0 and D15, D0 and D30 and D15 and D30 for normal distribution by paired Student’s t-test (*p* < 0.05) after using repeated ANOVA (*p* < 0.05), ^a,b,c^ Comparisons between D0 and D15, D0 and D30 and D15 and D30 for non-normal distribution by Wilcoxon signed rank test (*p* < 0.05) after using the Friedman test (*p* < 0.05). ^g,h,i^ Comparisons of Kruskal–Wallis test (*p* < 0.05) for four groups and Dunn test (*p* < 0.05) between two groups with non-normal distribution data. Abbreviations: Alb = albumin, ALT = alanine aminotransferase, AST = aspartate aminotransferase, Glo = globulin, TP = total protein.

## Data Availability

Not applicable.

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
