# Peer review of "Consumption of Sinlek Rice Drink Improved Red Cell Indices in Anemic Elderly Subjects"

_molecules, 2021, doi:10.3390/molecules26206285_

Round 1

Reviewer 1 Report

Table 1. authors present characteristics of the persons involved in the study; however, the more relevant data is respecting age and sex. Can the authors modify the information analysis, its table shows that these two variables have been influenced the results and effectiveness of Sinlek rice.

How affect the IR drinks and other variables involved in the study? In the group C is necessary to consider the mean of age and the number of females and Male on Hematopoietic Activity.

The authors do not show the significative effect on serum iron and total-iron capacity and either in the antioxidant capacity. The authors can be resalting the relation these with hematopoietic activity.

Author Response

Dear the reviewer,

Yours sincerely,

Professor Somdet Srichairatanakool, PhD.

Reviewer 2 Report

Thank you for submitting the manuscript “Consumption of Sinlek Rice Drink Improved Red Cell Indices in Anemic Elderly Subjects” to Molecules. The work is reasonably well written and some corrections are needed as highlighted in the PDF file. So, I even have few suggestions for this manuscript:
Line#23: what do the authors consider as low and high dose? Please insert dose in parentheses.
Line#28-31: “serum iron and transferrin saturation levels”, “serum antioxidant activity levels were increased” and “increase Hb and antioxidant capacity levels” it is necessary to say what this increase was and if it was significant. The abstract must bring numerical values to situate the reader of the article.
Line#45: “risk of mortality” is important to say because this mortality risk is increased.
Line#85: "In addition, the anemic subjects were found to be older than the non-anemic subjects (P < 0.05)." This is an important issue because the fact that this group is the leader may have influenced the positive result found in this group.
Tables: It is interesting to describe at the bottom of the table what group A, B, C and D mean, as tables and figures must always be self-sufficient with regard to information.
Lines#303-305: it is necessary to clarify whether the presence of hookworm in individuals has to do with low iron in the body or has to do with the consumption of rice or its derivatives contaminated by poor handling, lack of hygiene, lack of cooking, etc. As such, it appears that rice or its iron-rich derivatives cause more health problems than solve the low iron problem. The same is true for lines 347-349. Wasn't hookworm contamination a concern in the present manuscript?
Line#436: bee honey contains a large amount of macro and micronutrients, including iron. If the authors have the composition of this bee honey please include in addition to including the amount that was also used in the drink.

Author Response

Dear the Reviewer,

Yours sincerely,

Professor Somdet Srichairatanakool, PhD.,

Reviewer 3 Report

Reviewer's comment on Manuscript Number: molecules-1393088

The manuscript entitled “Consumption of Sinlek Rice Drink Improved Red Cell Indices 2 in Anemic Elderly Subjects” falls within the scope of Molecules.

The manuscript is certainly very interesting and valuable. This study could stimulate more research to further the exploration of the problem.  However, I would like to ask the authors to explain why vitamin C was not taken into consideration? It is the most important factor affecting the bioavailability of iron. Its level should be assessed both in volunteers as well as drinks. Besides, the authors stated in lines 119-120 that the increase in weight was surprising – could you explain this? I also propose to verify the manuscript by a native speaker.

I propose to accept this paper for publication in Molecules after minor amendments.

Author Response

Dear the Reviewer,

Yours sincerely,

Professor Somdet Srichairatanakool, PhD.
